# Effects of Replacing Alfalfa Hay with Oat Hay in Fermented Total Mixed Ration on Growth Performance and Rumen Microbiota in Lambs

**Mingjian Liu** [1,2,3,†] , **Yu Wang** [1,2,3,†], **Zhijun Wang** [1,2,3], **Gentu Ge** [1,2,3], **Yushan Jia** [1,2,3,*] **and Shuai Du** [4,5,6,7,8,*]

1   College of Grassland, Resources and Environment, Inner Mongolia Agricultural University, Hohhot 010019, China
2   Key Laboratory of Forage Cultivation, Processing and High Efficient Utilization, Ministry of Agriculture, Hohhot 010019, China
3   Key Laboratory of Grassland Resources, Ministry of Education, Hohhot 010019, China
4   National Engineering Laboratory of Biological Feed Safety and Pollution Prevention and Control, Hangzhou 310058, China
5   Key Laboratory of Molecular Nutrition, Ministry of Education, Hangzhou 310058, China
6   Key Laboratory of Animal Nutrition and Feed, Ministry of Agriculture and Rural Affairs, Hangzhou 310058, China
7   Key Laboratory of Animal Nutrition and Feed Science of Zhejiang Province, Institute of Feed Science, Hangzhou 310058, China
8   Zhejiang University, Hangzhou 310058, China
*   Correspondence: jys_nm@sina.com (Y.J.); nmgdushuai@zju.edu.cn (S.D.)
†   These authors contributed equally to this work.

**Abstract:** The use of the fermented total mixed ration (FTMR) is a promising approach for the preservation of feedstuff, but the effect of FTMR on the between growth performance and ruminal microflora of lambs are still limited. This study aimed to assess the effects of different roughage types in the FTMR on growth performance and rumen microbiota of lambs. Forty-five six-month-old Small tail Han sheep × Ujumqin male lambs were randomly allocated into three groups (three pens per treatment and five lambs per pen) with the initial body weight (BW) of 28.50 ± 1.50 kg. The three treatments were as follows: the low oat percentages group (LO) contained 200 g/kg oat hay + 400 g/kg alfalfa hay, the medium oat percentages group (MO) contained 300 g/kg oat hay + 300 g/kg alfalfa hay, and the high oat percentages group (HO) contained 400 g/kg oat hay + 200 g/kg alfalfa hay. The result revealed that the dry matter intake and average daily gain were markedly ($p < 0.05$) higher in the MO treatment than in the LO and HO treatments, whereas no significant difference ($p > 0.05$) was found in the final body weight. There were no significant ($p > 0.05$) differences on the Shannon and Simpson index among the three treatments. The PCoA score plot illustrated the individual separation in the LO, MO, and HO treatments. At the phylum level, the presence of *Bacteroidetes* and *Firmicutes* belonging to the dominant phyla is widely described in rumen communities among the three treatments. The relative abundances of *Prevotella*, *Fibrobacter*, and *Succinivibrio* in the level of the genes were remarkably higher ($p < 0.05$) in MO treatment than that in LO and HO treatments, while the relative abundance of *Sediminispirochaeta* was remarkably higher ($p < 0.05$) in LO treatment than that in MO and HO treatments. These results indicated that the MO treatments could more effectively improve growth performance than the LO and HO treatments, and also revealed that the different forage types in diets reshaped the compositions and function of the rumen microbiota. Consequently, the findings presented in this study provide a reference for the application of FTMR in animal production and the understanding of the interaction between diet, animal performance, and ruminal microbiota.

**Keywords:** oat; alfalfa; fermented total mixed ration; growth performance; rumen microbiota

## 1. Introduction

Roughage is a potential and important resource for the metabolism of ruminants. The rumen microorganisms could convert the roughage into volatile fatty acids (VFAs) and proteins to provide sufficient nutrients for ruminants by their unique physiological structure [1]. In fact, the fermentation process of rumen microbes is greatly affected by roughage sources [2]. Therefore, a better understanding of the interaction mechanism between roughage composition and rumen microbiota is helpful in increasing the productivity and economic efficiency of the animals.

Alfalfa (*Medicago sativa* L.) has been widely served as the primary source of roughage due to it being rich in protein and minerals [3]. However, the high cost of alfalfa hay production has severely restricted its availability in husbandry production [4]. Production practice shows that replacing alfalfa hay with other forages, including *Moringa oleifera* leaves [5], maize stover [6], and whole-plant corn [7], could be an effective method for reducing the drawbacks. Oat (*Avena sativa* L.) has been proven to be ideal for this popular feeding strategy due to the advantages of lower production cost and higher digestibility [8,9]. Zou et al. [10] demonstrated that portion utilization of oat hay in total mixed rations (TMR) could improve the ruminal function and growth performance of weaned calves. Notably, it has become common to apply total mixed ration to balance nutrition and improve nutrient utilization in animal production systems. A previous study illustrated that the use of the TMR feeding system was beneficial for the economic productivity and health of animals [11]. McAuliffe et al. [12] reported that the TMR treatment generally leads to higher milk production compared to the grazing system. However, the TMR feeding strategy usually results in higher labor costs due to the waste created by the sorting behavior of animals [13]. Consequently, the practice of mixing roughage with fermented total mixed ration (FTMR) may be an effective method to reduce feed waste [14,15]. Additionally, the previous report indicated that the FTMR is a combination of concentrate and roughage, formulated to fulfill a specific nutrient requirement of an animal [16]. Compared to the TMR feeding system, the FTMR feeding system could reduce the production of methane and optimize the rumen fermentation [17]. Meenongyai et al. [18] reported that the FTMR system could promote the ruminal fermentation and protein digestibility of Holstein-Zebu cross steers. Similarly, Zhang et al. [19] reported that the FTMR feeding system is a suitable choice for improving lactation performance and reducing the feed cost of dairy cows.

Rumen microbiota is a highly diverse but seriatim dynamic community. Inside this microbiome, bacteria are the dominant players as host to the digestion and conversion of feedstuffs which could provide more than 70% energy to fulfill the requirement of hosts [20,21]. The previous study demonstrated that diet composition, host genetics, feeding strategy, and other factors profoundly affected the construction of rumen microbiota [22]. Interestingly, the roughage sources have been commonly accepted as a potential target for manipulation to regulate ruminal microbiota metabolism and increase the growth performance of animals [23,24]. Zhu et al. [25] illustrated that different roughage diets could selectively modify the rumen bacterial colonization and benefit steer fatting. However, no studies have revealed the interaction mechanism between growth performance and microflora ruminal in FTMR containing different ratios of alfalfa and oat.

In the current study, the author hypothesized that replacing alfalfa hay with oat hay in the FTMR could provide suitable physical effective fiber in the diet and stimulate the production of cellulolytic bacteria in rumen to promote the growth performance of lamb. The aim of this study was to assess the effects of replacing alfalfa hay with oat hay in theFTMR on the growth performance and rumen microbiota of lambs.

## 2. Materials and Methods

### 2.1. Ethical Statement

The research protocol used in this study is based on the Institutional Guidelines for Animal Experiments of the Zhejiang University, Hangzhou, China. All experiments were performed according to the Institutional Animal Care and Use Committee at Zhejiang University (protocol #21905).

### 2.2. Preparation of FTMR

The alfalfa, oat, natural forage, corn stalk, and other raw material in the FTMR was obtained from the Chaoyue Feed Co., Ltd. (Balin Left Banner, Chifeng, China). The forage was shredded into lengths of 1–2 cm with a manual forage chopper and mixed with concentrate using a horizontal feed mixer (9JGM-9; Shijiazhuang Wantong Machinery Manufacturing Co. Ltd., Shijiazhuang, China). The moisture of the mixed materials was adjusted to 50% by spraying them with water. Meanwhile, a compound bacterial agent brought from Hebei Zhong bang Biotechnology Co., Ltd. (Strong brand, Cangzhou, China) was applied as fermented additives, which were dissolved in water and added at a ratio of 1 g/kg of fresh TMR. Thereafter, the FTMR was tightly filled into a special feed fermentation bag (55 cm × 85 cm) and then compacted and sealed in the bag mouth and stored indoors at 15 °C for approximately 60 days for fermentation.

### 2.3. Feed Composition Analysis

The dry matter (DM) content of the FTMR was calculated after oven-drying at 65 °C for 48 h until constant weight, and then being pulverized to pass through a 1 mm screen for the following analysis. The crude protein (CP) and organic matter (OM) were calculated by AOAC [26]. The water-soluble carbohydrate (WSC) content was analyzed using the anthrone colorimetry method [27]. Neutral detergent fibers (NDF) and acid detergent fibers (ADF) were measured using an ANKOM A200i fiber analyzer (ANKOM Technology) in accordance with Van Soest et al. [28]. Metabolizable energy (ME) was computed according to the method described by Freer et al. using the following formula: ME = GE − FE − UE − Eg [29].

To determine the fermentation characteristics of all FTMR samples, fermented samples of 10 g each were homogenized with a 90 mL sterile aqueous solution to extract the fermentation broth, and then four layers of gauze were used to filter the solution. The filtrate was serially diluted ($10^{-1}$ through $10^{-5}$) with sterilized water, then spread on agar plates so that the microorganisms could be enumerated. The de_Man Rogosa_Sharpe agar (MRS) was used to quantify the LAB numbers under anaerobic conditions, whereas the aerobic bacteria were counted by nutrient agar [30]. The pH value was assayed using a glass electrode pH meter (STARTER 100/B, OHAUS, Shanghai, China). To determine the content of organic acids, including lactic acid (LA), acetic acid (AA), propionic acid (PA), and butyric acid (BA), the filtrate was centrifuged at 12,000× $g$ for 10 min at 4 °C, and the supernatant was filtered through a 0.22 μm membrane filter. The filtrate was analyzed by high performance liquid chromatography (HPLC) with a UV detector (210 nm) and a column (e2695, Waters Co., Ltd., Milford, MA, USA). The mobile phase was 3 mmol $L^{-1}$ HClO4 at a flow rate of 1.0 mL $min^{-1}$ at 50 °C. The standard of organic was purchased from Dr. Ehrenstorfer GanbH. [31]. The ammonia nitrogen ($NH_3$-N) concentration was measured using the phenol-hypochlorite method according to the colorimetry [32].

### 2.4. Animals, Diets, and Experimental Design

The feeding experiment was carried out at Chaoyue Feed Co., Ltd. (Balin Left Banner, Chifeng, China). Forty-five six-month-old Small tail Han sheep × Ujumqin male lambs were randomly allocated into three groups (three pens per treatment and five lambs per pen) with initial body weight (BW) of 28.50 ± 1.50 kg. The low oat percentages group (LO) contained 200 g/kg oat hay + 400 g/kg alfalfa hay, the medium oat percentages group (MO) contained 300 g/kg oat hay + 300 g/kg alfalfa hay, and the high oat percentages group

(HO) contained 400 g/kg oat hay + 200 g/kg alfalfa hay. Based on the recommendation of NRC (2007) for lamb, the forage-to-concentrate ratio for three diet treatments was designed as 65:35 (DM basis); the ingredient compositions of the diet are illustrated in Table 1. The experiment lasted for 75 days, consisting of 15 days of an adaptation period and 60 days of a formal trial period. The lambs were fed twice daily with a feeding schedule at 07:00 and 17:00, allowing up to 10% orts. The lambs had free access to the trial diet and sufficient drinking water during the whole experimental period.

**Table 1.** Ingredients and chemical composition of dietary.

| Items | LO | MO | HO |
|---|---|---|---|
| Ingredient (g/kg DM) | | | |
| Oat hay | 200 | 300 | 400 |
| Alfalfa hay | 400 | 300 | 200 |
| Natural forage | 30 | 30 | 30 |
| Corn stalk | 20 | 20 | 20 |
| Corn | 220 | 200 | 180 |
| Soybean meal | 90 | 110 | 130 |
| Wheat bran | 20 | 20 | 20 |
| Calcium hydrogen phosphate | 3 | 3 | 3 |
| NaCl | 2 | 2 | 2 |
| $NaHCO_3$ | 5 | 5 | 5 |
| Premix | 10 | 10 | 10 |
| Chemical compositions | | | |
| DM (g/kg FW) | $45.73 \pm 0.46$ | $45.90 \pm 0.40$ | $45.77 \pm 0.49$ |
| CP (g/kg DM) | $13.33 \pm 0.12$ | $13.28 \pm 0.24$ | $13.12 \pm 0.17$ |
| NDF (g/kg DM) | $43.20 \pm 0.02$ | $50.29 \pm 0.01$ | $55.30 \pm 0.01$ |
| ADF (g/kg DM) | $29.53 \pm 0.02$ | $31.83 \pm 0.01$ | $33.77 \pm 0.01$ |
| WSC (g/kg DM) | $4.66 \pm 0.50$ | $4.91 \pm 0.52$ | $4.37 \pm 0.73$ |
| OM (g/kg DM) | $34.13 \pm 0.31$ | $34.65 \pm 0.46$ | $33.93 \pm 0.53$ |
| ME | $10.80 \pm 0.01$ | $10.70 \pm 0.01$ | $10.59 \pm 0.01$ |
| Fermentation profile | | | |
| pH | $4.55 \pm 0.06$ | $4.52 \pm 0.04$ | $4.49 \pm 0.02$ |
| Lactic acid (g/kg DM) | $10.28 \pm 0.08$ | $9.86 \pm 1.29$ | $9.69 \pm 0.93$ |
| Acetic acid (g/kg DM) | $1.15 \pm 0.01$ | $0.88 \pm 0.14$ | $0.76 \pm 0.18$ |
| Propionic acid (g/kg DM) | $1.24 \pm 0.01$ | $0.63 \pm 0.20$ | $0.94 \pm 0.14$ |
| Ammonia-N (g/kg DM) | $3.03 \pm 0.01$ | $2.65 \pm 0.16$ | $4.39 \pm 0.03$ |
| Microbial counts | | | |
| Lactic acid bacteria ($Log_{10}$ cfu/g FM) | $5.90 \pm 0.04$ | $7.81 \pm 0.01$ | $5.91 \pm 0.26$ |
| Aerobic bacteria ($Log_{10}$ cfu/g FM) | $6.05 \pm 0.18$ | $4.48 \pm 0.13$ | $6.74 \pm 0.50$ |

DM, dry matter; CP, crude protein; NDF, neutral detergent fiber; ADF, acid detergent fiber; WSC, water-soluble carbohydrate; OM, organic matter; ME, metabolizable energy; SEM, standard error of means. Composition of mineral premix. Per kg: Copper 1800 mg, iron 3400 mg, manganese 1500 mg, zinc 1700 mg, cobalt 20 mg, vitamin A 1,620,000 IU, vitamin D332 400 IU, vitamin E 540 IU, folic acid 15 mg. LO, low oat percentages group; MO, medium oat percentages group; HO, high oat percentages group.

*2.5. Feed Intake, Growth Performance, and Rumen Samples Collection*

The offered feed and refusals were measured and recorded daily to calculate the dry matter intake (DMI) throughout the trial. The BW of each lamb was weighed for each week between the beginning and end of the experiment to estimate the initial body weight (initial BW), final body weight (final BW), total weight gain (TWG), and average daily gain (ADG). At the end of the experiment, all lambs were transported to the slaughterhouse, where they were electrically stunned and slaughtered. Thereafter, the rumen samples were immediately gathered from the different sites of the rumen, and then they were filtered through four layers of cheesecloth to obtain the rumen fluid samples. Three rumen fluid samples were randomly selected for each treatment. Finally, a total of nine ruminal fluid samples (approximately 50 mL) were selected and immediately stored in liquid

nitrogen, and then preserved in a cryogenic refrigerator at −80 °C before analysis of 16S rRNA sequencing.

### 2.6. Bacterial DNA Extraction, Polymerase Chain Reaction Amplification, and 16S rDNA Sequencing

Total DNA from rumen fluid samples were extracted with a commercial sample with E.Z.N.A. ®Stool DNA Kit (D4015, Omega, Inc., Norwalk, CT, USA) using a modification of the procedure. The agarose gel electrophoresis (2%) and NanoDrop 2000 UV–vis Spectrophotometer (Thermo Scientific, Wilmington, DE, USA) were used to determine the concentration and purity of extracted DNA according to the description of Ma et al. [33]. Primers targeting the V3–V4 regions of 16S rDNA 341 (5′- CCTACGGGNGGCWGCAG-3′) and 805 (5′-GACTACHVGGGTATCTAATCC-3′) were selected to conduct PCR amplification [34], and which were performed by LC-Bio Technology Co., Ltd. (Hangzhou, China). Sequencing data for 16S rRNA gene sequence were stored in NCBI with BioProject accession number PRJNA899538.

### 2.7. Bioinformatics Analysis

Purified DNA was determined by Paired-end sequencing and the Illumina MiSeq PE300 platform (Illumina Inc., San Diego, CA, USA). Thereafter, the low-quality reads were screened and trimmed to obtain high-quality clean reads using fqtrim (v0.94), after which the chimeric sequences were filtered through Vsearch (v2.3.4) [35]. After dereplication using DADA2, the feature table and sequence were determined [36]. The high-quality clean sequences with a threshold of 97% similarity were assigned to the same operational taxonomic units (OTUs) [37]. The bioinformatics data were examined via the free online platform at https://www.omicstudio.cn/index (accessed on 1 November 2022). The QIIME (version 1.9.1) was applied to estimate the complexity of species diversity via alpha diversity indices (Chao1, Shannon, and Simpson), and Good's coverage analysis was also performed. The principal coordinate analysis (PCoA) with weight-Unifrac distance metric was constructed by R (version 1.7.13) [38] to calculate the β-diversity. The common and unique OTUs were performed to plot the Venn diagram by R (version 1.6.20) [36]. The linear discrimination analysis (LDA) combined with effect size (LEFSe) was analyzed to assess the primary differentially abundant genera generated from different dietary treatments [39]. The 16S sequencing data were exported into the Kyoto Encyclopedia of Genes and Genomes (KEGG) database to infer the rumen microbiota functional pathways using PICRUSt2 [40]. Bar plots were presented and plotted with GraphPad Prism 9 (San Diego, CA, USA).

### 2.8. Statistical Analysis

Data on the growth performance of lambs (Initial BW, Final BW, TWG, ADG, DMI) and diversity indices of ruminal microbiota (OTUs, Chao1, Simpson, Shannon, Goods' coverage) were analyzed using a one-way analysis procedure of the SAS ver. 9.2 according to the statistical model: $Y = \mu + \alpha + \varepsilon$, where Y = observation, $\mu$ = overall mean, $\alpha$ = diet effect and $\varepsilon$ = error. Duncan's tests separated significant differences, and $p < 0.05$ was taken as statistical significance. The data values of the experiment are represented as the mean of triplicate measurements among different treatments and the standard error of the mean (SEM).

## 3. Results

### 3.1. Animal Performance

The growth performance of lambs fed FTMRs with different ratios of alfalfa and oat are presented in Table 2. As expected, the initial weights were similar among the LO, MO, and HO treatments. There was no significant ($p > 0.05$) difference in Final BW among the three treatments observed. Compared to the LO and HO treatments, the highest TWG (16.17 kg) and ADG (269.44 g/day) were detected in the MO treatment. Notably, The

DMI content was significantly greater ($p < 0.05$) in the MO treatment than in the LO and HO treatments.

**Table 2.** Growth performance of lambs fed fermented total mixed rations with different ratio of alfalfa and oat.

| Items | LO | MO | HO | SEM | *p*-Value |
|---|---|---|---|---|---|
| Initial BW (kg) | 28.00 | 29.33 | 29.17 | 0.333 | 0.2205 |
| Final BW (kg) | 42.83 | 45.50 | 42.17 | 0.764 | 0.1715 |
| TWG (kg) | 14.83 ab | 16.17 a | 13.00 b | 0.601 | 0.0423 |
| ADG (g/day) | 247.22 ab | 269.44 a | 216.67 b | 10.015 | 0.0423 |
| DMI (kg/day) | 1.17 b | 1.18 a | 1.13 c | 0.008 | <0.0001 |

BW, body weight; TWG, total weight gain; ADG, average daily gain; DMI, dry matter intake; SEM, standard error of the mean. Means within the same rows with different letters are significantly different ($p < 0.05$). SEM, standard error of means. LO, low oat percentages group; MO, medium oat percentages group; HO, high oat percentages group.

*3.2. Rumen Bacterial Communities*

The alpha diversity indices of lambs fed FTMRs with different ratios of alfalfa and oat are illustrated in Table 3. Interestingly, no significant ($p > 0.05$) differences were found in OTUs, Chao1, Simpson, and Shannon indexes among the three treatments. Similarly, the Goods' coverage in the LO, MO, and HO treatments was more than 99%, indicating that the actual situation of the sampling was enough to investigate the accuracy and reproducibility within the dataset.

**Table 3.** Diversity indices of ruminal microbiota of lambs.

| Items | LO | MO | HO | SEM | *p*-Value |
|---|---|---|---|---|---|
| OTUs | 2446 | 2276 | 2627 | 110.723 | 0.5390 |
| Chao1 | 2448.61 | 2282.45 | 2633.22 | 110.220 | 0.5359 |
| Simpson | 1.00 | 0.99 | 0.97 | 0.007 | 0.1790 |
| Shannon | 9.48 | 8.98 | 8.68 | 0.192 | 0.3059 |
| Goods' coverage | 99.88 | 99.92 | 99.93 | 0.001 | 0.5582 |

Means within the same rows with different letters are significantly different ($p < 0.05$). SEM, standard error of means. LO, low oat percentages group; MO, medium oat percentages group; HO, high oat percentages group.

Overall, 753,662 raw reads were obtained. Based on a 97% sequence identity threshold, a total of 18,862 OTUs were estimated in the LO, MO, and HO treatments, with an average of 83,740 sequences per rumen sample. Of these, 846 OTUs were shared in all treatments, while 4396, 4313, and 5267 OTUs were exclusive to the LO, MO, and HO treatments, respectively (Figure 1A).

The PCoA was plotted based on the weighted UniFrac distance metric to further estimate the effects of lambs fed FTMRs with different ratios of alfalfa and oat on differences in bacterial composition (Figure 1B). The result of the present research indicated that the bacterial community structures in the LO, MO, and HO treatments separated effectively from each other.

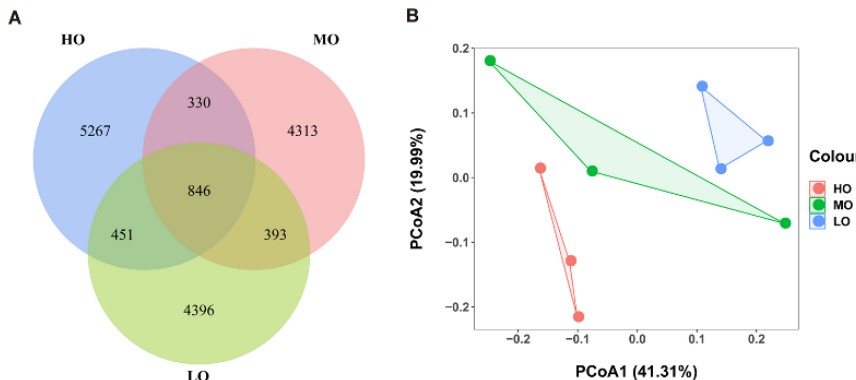

**Figure 1.** Microbial community among different treatments (*n* = 3). (**A**) Venn diagram representing the common and unique operational taxonomic units (OTUs) found at each treatment. (**B**) Principal coordinates analysis (PCoA) of samples conducted based on weighted UniFrac distance. LO, low oat percentages group; MO, medium oat percentages group; HO, high oat percentages group.

The result of taxonomic analysis indicated that a total of 25 bacterial phyla and 414 bacteria genera were detected in the rumen samples. At the phylum level, the relative abundance of *Bacteroidetes*, *Firmicutes*, *Kiritimatiellaeota*, *Spirochaetes*, *Proteobacteria*, *Fibrobacteres*, and *unclassified* were more than 1%. They were served as the most abundant phyla in the LO, MO, and HO treatments (Figure 2A). Compared with the LO and HO treatments, the phylum *Fibrobacteres* was significantly ($p > 0.05$) higher in the MO treatment (Figure 2B). At the genus level, the main genera included *Prevotella_1*, *Bacteroidales_RF16_group_unclassified*, *WCHB1-41_unclassified*, *Rikenellaceae_RC9_gut_group*, and *F082_unclassified* (Figure 2C). The relative abundances of *Prevotella*, *Fibrobacter*, and *Succinivibrio* were remarkably higher ($p < 0.05$) in the MO treatment than in the LO and HO treatments. Compared with the MO and HO treatments, the relative abundance of *Sediminispirochaeta* was remarkably higher ($p < 0.05$) in the LO treatment. Notably, the HO treatment remarkably increased the relative abundances of *Pseudomonas* (Figure 2D).

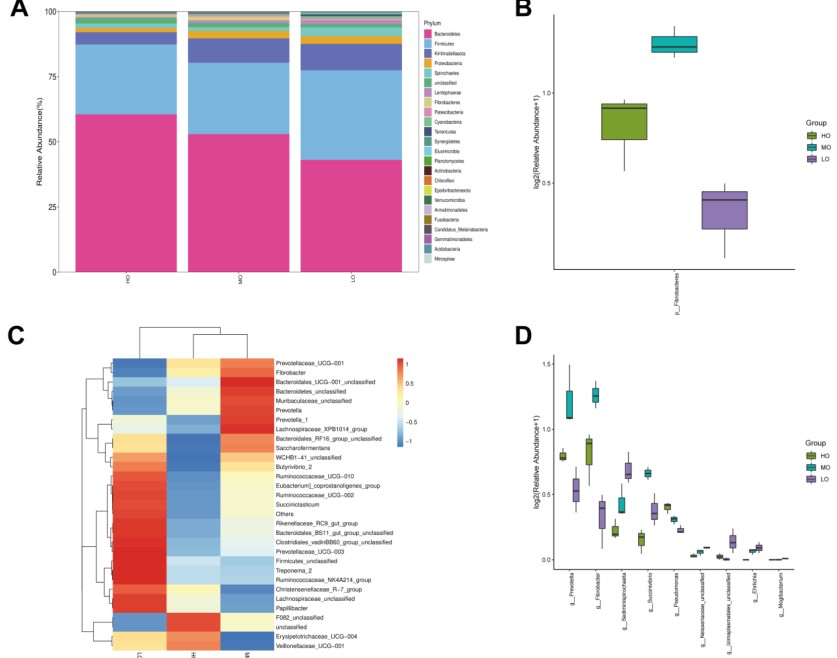

**Figure 2.** The relative abundance (%) of bacterial phyla (Top 30) of ruminal microbiome of lambs fed FTMR with different alfalfa and oat ratio (*n* = 3). (**A**) Phylum level. (**B**) Extended error bar plot showing the bacteria at the phylum level that had significant differences among the LO, MO, and HO

groups. (**C**) Genus level. (**D**) Extended error bar plot showing the bacteria at the genus level that had significant differences among the LO, MO, and HO groups. LO, low oat percentages group; MO, medium oat percentages group; HO, high oat percentages group.

The LEfSe analysis was performed to reflect the variations of the differences in bacterial community structures at various taxonomic levels of the LO, MO, and HO treatments (LDA score > 3.0) (Figure 3). In the present research, the result revealed that the *Fibrobacter*, *Prevotella*, and *Succinivibrio* were mainly enriched in the MO treatment. In contrast, the *Sediminispirochaeta* was enriched peimarily in the LO treatment. Notably, no remarkable ($p > 0.05$) differences in bacterial community structure were found in the HO treatment related to the MO and LO treatments.

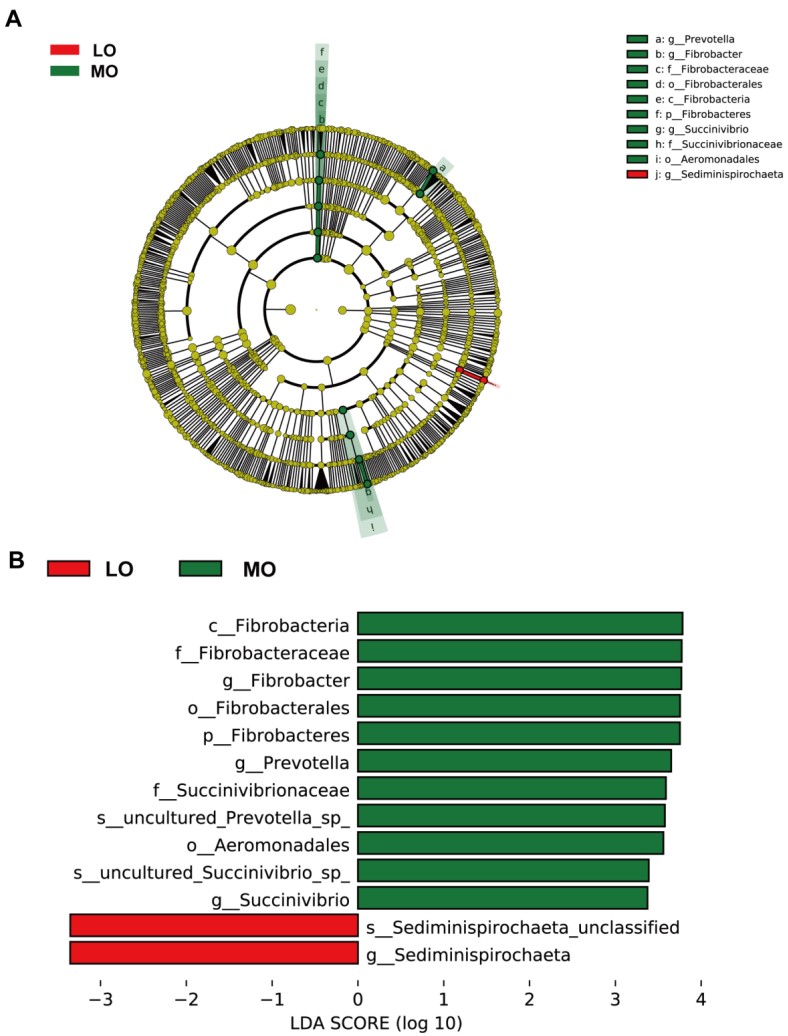

**Figure 3.** Linear discrimination analysis (LDA) coupled with effect size (LEfSe) analysis of the rumen microbial community of lamb in the LO, MO, and HO groups (*n* = 3). (**A**) Cladogram showing microbial species with significant differences among the two treatments. Red and green represent different groups. Species classification at the phylum, class, order, family, and genus level are displayed from inner to outer layers. The red and green nodes represent microbial species in the phylogenetic tree that play important roles in the MO and LO groups, respectively. Yellow nodes represent no significant difference between species. (**B**) Significantly different species with an LDA score greater than the estimated value (default score = 3). The length of the histogram represents the LDA score of different species in the LO, MO, and HO groups. LO, low oat percentages group; MO, medium oat percentages group; HO, high oat percentages group.

### 3.3. Predicted Metabolic Pathways and Functions of Rumen Bacterial Communities

PICRUSt2 was selected to predict the microbial function and pathway profile of rumen bacterial communities (Figure 4). The result showed that the main predicted functional genes at level 1 in LO, MO, and HO treatments were much associated with metabolism (48.67–49.48%), genetic information processing (22.93–23.48%) and environmental information processing (9.32–10.46%), respectively (Figure 4A). At KEGG level 2, genes belonging to replication and repair, translation, amino acid metabolism, carbohydrate metabolism, energy metabolism, nucleotide metabolism, and membrane transport accounted for more than 5% of the enriched pathways among the three treatments (Figure 4B). Among them, the amino acid metabolism and carbohydrate metabolism were markedly ($p < 0.05$) increased in the MO treatment. At the 3 levels, some differences in the majority pathways were observed in the three treatments (Figure 4C). The abundance of a majority of the genes related to the two-component system, ABC transporters, transporters, pyruvate metabolism, and transcription factors was increased dramatically ($p < 0.05$) in the LO treatment. Interestingly, the genes associated with carbon fixation pathways in prokaryotes, amino acid-related enzymes, peptidases, pyrimidine metabolism, purine metabolism, homologous recombination, DNA replication proteins, and DNA repair and recombination proteins were markedly ($p < 0.05$) increased in the HO treatment. Additionally, the genes related to oxidative phosphorylation, transcription machinery, and chromosome were markedly ($p < 0.05$) assigned to the MO treatment.

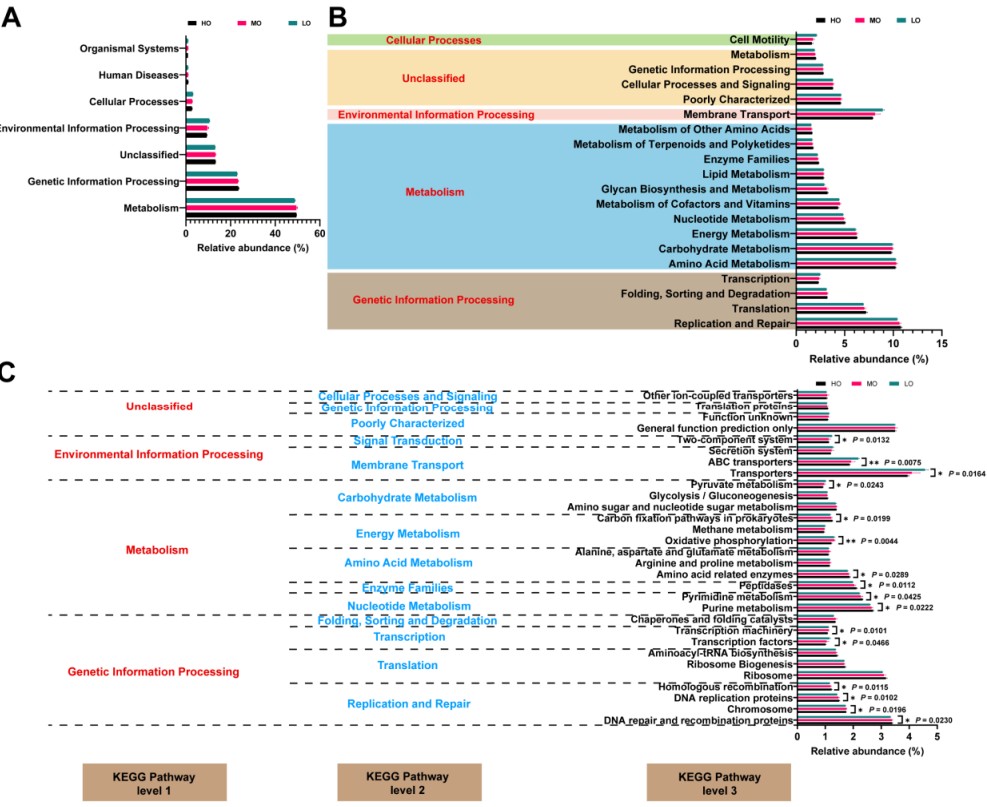

**Figure 4.** Dynamics of rumen bacterial predicted functional profiles fed with different diets analyzed by PICRUSt2 ($n = 3$). (**A**) Level 1 metabolic pathways. (**B**) Level 2 Kyoto Encyclopedia of Genes and Genomes (KEGG) ortholog functional predictions of the relative abundances of the top 20 metabolic functions. (**C**) Level 3 KEGG ortholog functional predictions of the relative abundances of the top 30 metabolic functions. LO, low oat percentages group; MO, medium oat percentages group; HO, high oat percentages group.

## 4. Discussion

An appropriate composition of roughage is the key to improving rumen microbiota and the rapid growth of ruminants [41]. This study characterized the effect of replacing alfalfa with oat in FTMRs on growth performance and rumen microbiota in lambs using 16S rRNA gene sequences, which provide a reference in understanding the relationship between the animal performance, rumen microbiota, and roughage composition.

The results of the present research revealed that the DMI of lambs was markedly increased in MO treatment than in LO and HO treatments, indicating that the intake of lambs was affected by the FTMR, following prior research that the fermented feeding system could increase the feed intake of cattle [42], which may be attributed to differences in smells and taste caused by various organic acids among the three treatments leading to the choice behavior of feed and the change of palatability of lambs [43–45]. Similarly, Cho et al. [46] demonstrated that the feed intake of Hanwoo steers could be promoted by the supplementation of fermented feed. Generally, the growth rates of animals are determined mainly by the intake of nutrients and energy [47], which in the present research indicated that the higher DMI might be the main reason for the increase of ADG and TWG in the MO treatment than in the LO and HO treatments [20]. Additionally, the adequate amount of lactic acid bacteria in the diet may improve growth performance by increasing the abundances of *Prevotella* and *Succinivibrio* [48]. Therefore, the higher ADG in the MO treatment may be further explained by the higher lactic acid bacteria numbers in the MO treatment.

It is well known that diets could regulate the rumen microbiota and affect the feed utilization rate [49]. In the present study, no significant difference was found in the alpha diversity among the three treatments. In contrast, the results of the PCoA illustrated that the bacterial community structures in the LO, MO, and HO treatments separated effectively from each other. These results indicated that the forage type in the FTMR did not affect the richness of the rumen microbiota while reshaping the microbiota composition of lambs, which was in agreement with Wang et al. [50], who reported that the forage type had no adverse effects on the alpha diversity of ruminal bacteria while having a considerable influence on the microbiota composition. Similarly, these results could be commonly validated by previous studies in ruminants [51,52], and the preference of different bacteria to grow in various rumen environments and nutritional ingredients may be the main reason [53,54].

Ruminal microorganisms were tightly correlated to ruminant growth and metabolism [55]. At the phylum level, the presence of *Bacteroidetes* and *Firmicutes* belonging to the dominant phyla was widely described in rumen communities, which followed the previously published studies wherein *Bacteroidetes*, *Firmicutes*, *Proteobacteria*, and *Kiritimatiellaeota* were identified as the phylum-level core microbiomes and account for more than 90% of bacterial species [56,57]. The *Bacteroidetes* and *Firmicutes* were tightly correlated to the degradation of plant polysaccharides and fiber. In this study, the *Bacteroidetes* and *Firmicutes* were the two most abundant bacterial phylum, and they were stable in the HO, MO, and LO treatments, which may reflect the presence of the core microbiome [58]. The *Fibrobacteres* could provide nutrients for ruminants by manipulating the process of degradation of fiber and cellulose. The MO treatment had a remarkably increased relative abundance of *Fibrobacteres* compared to the LO and HO treatments, which may be due to the digestion of dietary fiber existing the intermediate disturbance and specialized niches [56,59]. Furthermore, LAB have been shown to exert probiotic effects in promoting ruminant performance. In the current study, the higher amount of lactic acid bacteria in the MO treatment may have increased the fermentation of metabolites to promote the production of cellulolytic bacteria, which could increase the growth performance of animals [60]. Similarly, Wang et al. [48] reported that the adequate amount of lactic acid bacteria in the diet enhanced the rumen microbial population (especially the cellulolytic population).

At the genus level, the effects of different forage types in FTMRs on the rumen bacterial population were further identified in this study. Notably, the release of hormones and nutrients into the tissues of animals was tightly correlated to the production of propionic acid in the rumen [61,62]. Generally, most of the propionate in the rumen was produced by the succinate-producing bacteria and the succinate-to-propionate-reducing bacteria [63]. The *Prevotella* could break down and utilize carbohydrates, which was recognized to be correlated with the production of acetate and succinic acids as the end products of fermentation [64,65]. Members of the *Fibrobacter* genera produce succinate as one of its products and play a crucial role in fiber digestion in the rumen [66]. In the present study, the relative abundances of *Prevotella* and *Fibrobacter* were remarkably higher ($p < 0.05$) in the MO treatment than in the LO and HO treatments, indicating that an appropriate composition of roughage in the FTMR improved the digestion ability of lambs, which may be explained by the higher number of lactic acid bacteria in the MO treatment after fermentation. The lactic acid bacteria could contribute to the hydrolysis of structural carbohydrates in the rumen, which may have stimulated the proliferation of cellulolytic bacteria and the higher propionate generation [67–69]. Taken together, an increase in *Prevotella* and *Fibrobacter* in the MO treatment confirms its essential role in utilizing the carbohydrate and promoting the production of propionic acid in ruminants and at least partially explains the improvement in ADG. The *Succinivibrio* was specialized in hemicellulose and structural carbohydrates degradation [70]. In this study, the MO treatment had a remarkably increased abundance of *Succinivibrio*, which may be interpreted as a need for more specialized fermentation due to the more nutrient intake caused by the higher DMI in MO treatment [70,71]. The previous study has reported that *Sediminispirochaeta* relates to the production of acetate [72]. Compared with the MO and HO treatments, the relative abundances of *Sediminispirochaeta* was remarkably higher ($p < 0.05$) in the LO treatment, which may be because the higher acetate proportion induced an increase in acetate proportion in the rumen, thus increasing the abundance of *Sediminispirochaeta* [73,74].

The rumen microbial functions were tightly related to the dietary composition. Our results revealed that several predicted KEGG pathways from level 1 to 3 were generally consistent. Compared with the LO and HO treatments, genes involved in carbohydrate metabolism and amino acid metabolism were increased in lambs in the MO treatment. This result could be attributed to the higher abundance of *Prevotella* and *Fibrobacter* in the MO treatment, which could eventually degrade protein and hemicellulose to promote the effective absorption of nutrients [75,76]. Similarly, Cui et al. [55] affirmed the functions of cellulolytic bacteria in regulating the carbohydrate metabolism and amino acid metabolism. Additionally, the metabolism of cofactors and vitamins was extremely enriched inthe MO treatment compared to the LO and HO treatments, which is in agreement with Yildirim et al. [77], who reported that the dietary composition affected pathways of the metabolism of vitamins by shaping the rumen microbiota. The result of the present study may be explained by the higher abundance of microorganisms involved in fiber decomposition, which could digest cellulose accompanied with synthesized vitamins [78].

## 5. Conclusions

Compared with the LO and HO treatments, the DMI and ADG of lambs were remarkably increased in the MO treatment. Additionally, the result revealed that the interaction between diet and ruminal microbiota reshaped the compositions and function of the rumen microbiota in the MO treatment, and resulted in the elevated production performance of lambs. The absence of rumen fermentation parameters was one of the limitations of this study. Regardless, the results of the present study can provide a reference for the application of FTMRs in animal production and the understanding of the interaction between diet, animal performance, and ruminal microbiota.

**Author Contributions:** Conceptualization, methodology, data curation, writing-original draft preparation and writing-review and editing; M.L. Methodology: Y.W. Investigation and resources: Z.W. Writing—Review and editing; G.G. Project administration and funding acquisition: Y.J. Writing-original draft preparation, Writing—Review and editing; S.D. All authors have read and agreed to the published version of the manuscript.

**Funding:** This research was financially supported by the National Dairy Technology Innovation Center Creates Key Projects (2021-National Dairy Centre-1).

**Informed Consent Statement:** Not applicable.

**Data Availability Statement:** Sequencing data for 16S rRNA gene sequence were stored in NCBI with BioProject accession number PRJNA899538.

**Acknowledgments:** We thank Chaoyue Feed Co., Ltd. (Balin Left Banner, Chifeng, China) for providing the lambs.

**Conflicts of Interest:** The authors declare no conflict of interest.

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
