# Peer review of "Effects of Replacing Alfalfa Hay with Oat Hay in Fermented Total Mixed Ration on Growth Performance and Rumen Microbiota in Lambs"

_fermentation, doi:10.3390/fermentation9010009_

Round 1

Reviewer 1 Report

In this study the authors investigated the effect of replacing alfalfa with oat in diets for lambs on growth performance and rumen microbiota in lambs. The presented results are interested as for readers dealing with nutrition and farming of ruminants but also for academia, since the results of rumen microbiota gives insight in area that can be a field of future researches. The layout of the paper is suitable.  The given methodology is well detailed. The discussion is clear and easy to follow. The English requires some improvements, especially in abstract and the introduction section. I would recommend this paper for publishing if author manage to make several corrections:

Please change the term middle oat percentage group into MEDIUM oat percentage group throughout the text.

Lines 41 – 44: Please rephrase the sentence. The meaning of current sentence is that nutrients are converted by rumen microbiota into volatile fatty acids and proteins in order to provide nutrients, which is confusing.

Author Response

Dear Editor and Reviewers,

Thank you very much for evaluating our paper.

We thank you very much for giving us the opportunity to revise our manuscript. We appreciate editor and reviewers very much for their positive and constructive comments and suggestions on our manuscript entitled “Effects of Replacing Alfalfa Hay with Oat Hay in Fermented Total Mixed Ration on Growth Performance and Rumen Microbiota in Lambs” (fermentation-2074425). The comments and suggestions are not only helpful for us to revise and improve our manuscript, but also benefit our further research. We hope that our paper much better quality than before.

Best regards,

Dr. Mingjian Liu,

Dr. Yushan Jia                                                      Dr. Shuai Du

E-mail: [email protected]                                  E-mail: [email protected]

Reviewer 2 Report

This manuscript needs to be reorganized in some parts of the material and methods, the hypotheses have to be inserted clearly.

In the abstract, a short introduction is needed before the aim of the study.

Line 64, please insert a short description of the FTMR, and suitable references that compared the TMR with FTMR feeding system.

Line 80 and the further paragraph seem to e not linked together to add the word "therefore", in your hypothesis it is not clear why the authors assumed that replacing alfalfa hay with oat hay in FTMR would promote the growth performance. explanations have to be inserted here.

Line 98, the recommendations of NRC for which animal?

Line 100, during the 15-day pre-trial, what was the basic feeding system? is it the adaptation period? is so insert the basal diet.

Line 101, the animals were fed twice a day their diets, which diet? are these dietary treatments?

This part "Preparation of FTMR 110" has to be the first part of the material and methods, then the part of "Feed Composition Analysis" and the third part will be the "Animals, diets, and experimental design"

     Line 102, what was the basic diet?

Line 141, please insert the source of the standards,  conditions, model, city, and country of the GC used for the VFA determination.

Line 142, colorimetrically?

Line 152, from just three animals or rumens?

write the full name of WSC in table 1!

For the statistical analysis, you just referred to how you analysied the growth prformnace data!! so what about the other experimental parameters, and how they were analyzed? what was the experimental unit, and the number of the statical repetitions for the experimental parameters?

Numbers in all tables have to be in three digits.

in the statics part, you said that you used the SEM, while in table 1 you used SE, i suggest using the SEM to obtain clear and not crowded tables.

Please remove the first sentence in the conclusions, the aim of the study has to be only in the introduction, just the most important results have to be inserted in the conclusions.

Author Response

(The authors gave the same response as above.)
